# Global health security and universal health coverage: Understanding convergences and divergences for a synergistic response

Yibeltal Assefa[1]*, Peter S. Hill[1], Charles F. Gilks[1], Wim Van Damme[2], Remco van de Pas[2], Solomon Woldeyohannes[1], Simon Reid[1]

1 School of Public Health, the University of Queensland, Brisbane, Australia, 2 Institute of Tropical Medicine, Antwerp, Belgium

* y.alemu@uq.edu.au

## Abstract

### Background

Global health security (GHS) and universal health coverage (UHC) are key global health agendas which aspire for a healthier and safer world. However, there are tensions between GHS and UHC strategy and implementation. The objective of this study was to assess the relationship between GHS and UHC using two recent quantitative indices.

### Methods

We conducted a macro-analysis to determine the presence of relationship between GHS index (GHSI) and UHC index (UHCI). We calculated Pearson's correlation coefficient and the coefficient of determination. Analyses were performed using IBM SPSS Statistics Version 25 with a 95% level of confidence.

### Findings

There is a moderate and significant relationship between GHSI and UHCI ($r = 0.662$, $p<0.001$) and individual indices of UHCI (maternal and child health and infectious diseases: $r = 0.623$ ($p<0.001$) and $0.594$ ($p<0.001$), respectively). However, there is no relationship between GHSI and the non-communicable diseases (NCDs) index ($r = 0.063$, $p>0.05$). The risk of GHS threats a significant and negative correlation with the capacity for GHS ($r = -0.604$, $p<0.001$) and the capacity for UHC ($r = -0.792$, $p<0.001$).

### Conclusion

The aspiration for GHS will not be realized without UHC; hence, the tension between these two global health agendas should be transformed into a synergistic solution. We argue that strengthening the health systems, in tandem with the principles of primary health care, and implementing a "One Health" approach will progressively enable countries to achieve both UHC and GHS towards a healthier and safer world that everyone aspires to live in.

**Data Availability Statement:** The minimal data set underlying the results described in the manuscript can be found at: Global Health Security Index 2019:

https://www.ghsindex.org/ and World Health Organization: https://www.who.int/data/gho/data/themes/universal-health-coverage.

**Funding:** The research was funded by the University of Queensland. The University had no role in study design, data collection and analysis, decision to publish, or preparation of the manuscript.

**Competing interests:** The authors have declared that no competing interests exist.

## Background

The pandemic of Coronavirus Disease 2019 (COVID-19) is a timely reminder of the nature and impact of emerging infectious diseases that are public health emergencies (PHEs) of international concern [1]. Public health emergencies are defined as "occurrence or imminent threat of an illness or health condition caused by epidemic or pandemic disease, bio terrorism, or (a) novel and highly fatal infectious agent or biological toxin, that poses a substantial risk of a significant number of human fatalities or incidents or permanent or long-term disability" [2]. They have increasingly captured the attention of global health and security communities, and become an emergent theme in recent academic and policy discourses [3, 4]. Various approaches, strategies and programs have been developed to address PHEs at national and global levels, including initiatives to strengthen public health preparedness and global health security (GHS), which represent the proactive and reactive efforts required to protect the world's population from PHEs [5].

The key global strategy to achieve GHS is implementation of the International Health Regulations (IHR), which is the international law on public health that provides guidance on how human societies should govern their vulnerabilities to PHEs [6]. International Health Regulations require all countries to improve capacity in prevention, early detection, and timely and effective response to the international spread of disease [7]. The 2014 Ebola virus disease outbreak in west Africa brought renewed attention to GHS towards strengthening core public health capacities according to IHR (2005) [3, 8]. Therefore, the GHS Agenda (GHSA) came about as an international collaborative effort among governments and across sectors to strengthen the implementation of the IHR (2005) [9]. The GHSA is based on a "One Health" approach to health security [10], recognizing that the health of people is connected to the health of animals and the environment. Nearly two-thirds of known pathogens and three-quarters of newly emerging pathogens spread from animals to humans [11].

In 2018, the World Health Organization (WHO) released its 13[th] general programme of work, which aims to promote health, keep the world safe, serve the vulnerable, sets out priorities towards ensuring healthy lives and promoting well-being for all at all ages by: achieving universal health coverage (UHC)–one billion more people benefitting from UHC; addressing health emergencies—one billion more people better protected from health emergencies; and promoting healthier populations—one billion more people enjoying better health and well-being [12]. These strategic priorities require implementation that is jointly reinforcing [13].

However, there is insufficient reconciliation [14], and tension between GHS and UHC conceptualisation, strategy and implementation [15, 16]. In addition, there are disagreements within the broader GHS community on security from what kind of health threat and for whom [4]. In this situation, one should ask: how can we establish and strengthen synergistic systems that will provide health security to everyone [17]?

The objective of this study is to assess the relationship between GHS and UHC by determining their correlation using two recent quantitative indices. Currently, UHC is monitored by the UHC index (UHCI) [18, 19] and GHS by the GHS index (GHSI) [20]. By doing that, we will identify how countries and the global health community can achieve both GHS and UHC (without undermining the other) towards a healthier and safer world.

## Methods

### Data sources

The GHS index (GHSI, the John Hopkins University) is a composite measure to assess a country's capability to prevent, detect, and respond to epidemics and pandemics. It is calculated using a framework based on 140 questions organized across six categories: (1) Prevention

(prevention of the emergence or release of pathogens); (2) Detection and reporting (early detection and reporting for epidemics of potential international concern); (3) Rapid response (rapid response to and mitigation of the spread of an epidemic); (4) Health system (sufficient and robust health system to treat the sick and protect health workers); (5) Compliance with international norms (commitments to improving national capacity, financing plans to address gaps, and adhering to global norms); and, (6) Risk environment (overall risk environment and country vulnerability to biological threats) [20]. The GHSI is calculated using 140 questions, 34 indicators and 85 sub-indicators, which rely entirely on open-source data from countries. The overall score (0–100) for each country is a weighted sum of the category scores (prevention (16.3%), detection (19.2%), response (19.2%), health systems (16.7%), compliance (15.8%) and risk (12.8%)) assigned by International Panel of Experts. A score of 100 does not indicate that a country has perfect national health security conditions; likewise, a score of 0 does not mean that a country has no capacity. Scores of 0 and 100 represent the least and the most, respectively, favourable health security conditions [20].

The UHC services coverage index (UHCI, WHO) is calculated as the geometric mean of the coverage of essential services based on 17 tracer indicators from the following four categories: (1) reproductive, maternal, newborn and child health; (2) infectious diseases; (3) non-communicable diseases; and, (4) service capacity and access and health security. All indicators are structured so they occur on a scale of 0 to 100%. The UHCI is constructed from geometric means of component indicators in two steps: first, within each of the four categories, and then across those category-specific means. A simple equal weighting approach was utilized when computing the index [18]. Due to data limitations, not all tracer indicators used to compute the index are direct measures of service coverage. Hence, the selected tracer indicators are meant to represent the broad range of essential health services necessary to monitor the progress towards UHC [19].

## Data analysis

We prepared a table consisting of the indictors for UHCI in the first column and for GHSI in the second column with the intention to identify areas of overlap and non-overlap of the different indicators for these two indices. We summarized the UHCI and the GHSI data for the six regions of the world, using Microsoft Excel Sheet 2016, to identify the levels of and differences in UHCI and GHSI across the different regions of the world. We calculated Pearson's correlation coefficient to assess the relationship between the UHCI and GHSI for all included countries (183 in both indices) and the coefficient of determination to estimate how much UHCI can explain the variability in GHSI. List-wise deletion is used to manage the missing data. Analyses were performed using IBM SPSS Statistics Version 25 with a 95% level of confidence.

The findings of the study are structured and presented in order to: (1) identify the presence of convergence and/or divergence between UHC and GHS, and (2) assess GHS "for whom" and "from what". First, we will describe the concordance and discordance of the indicators for UHCI and GHSI. Second, we will analyse the level and equity of UHCI and GHSI across the six WHO regions. Third, we will assess the relationship between UHCI and GHSI. Finally, we will identify "for whom" and "from what" GHS is designed to provide security.

## Findings

### Concordance of indicators for universal health coverage and global health security

Table 1 presents the indicators for UHC and GHS. The table demonstrates that there is substantial concordance (overlap) in the indicators used to calculate UHCI and GHSI. There are

**Table 1. Mapping universal health coverage and global health security indicators: Areas of concordance and discordance\*.**

| Universal Health Coverage | Global Health Security | | | | | |
|---|---|---|---|---|---|---|
| **Reproductive, maternal, newborn and child health:** <br> • Family planning <br> • Antenatal and delivery care <br> • Full child immunization <br> • Health-seeking behaviour for pneumonia. <br> **Infectious diseases:** <br> • Tuberculosis treatment <br> • HIV antiretroviral treatment <br> • Hepatitis treatment <br> • Use of insecticide-treated bed nets for malaria prevention <br> • Adequate sanitation <br> **Non-communicable diseases:** <br> • Prevention and treatment of raised blood pressure <br> • Prevention and treatment of raised blood glucose <br> • Cervical cancer screening <br> • Tobacco (non-)smoking. <br> **Service capacity and access:** <br> • Basic hospital access <br> • Health worker density <br> • Access to essential medicines <br> • Health security: compliance with the international health regulations. | **Prevention:** assess antimicrobial resistance, zoonotic disease, biosecurity, dual – use research and culture of responsible science, and immunization. | **Detection and reporting:** assess laboratory systems; real – time surveillance and reporting; epidemiology workforce; and data integration between the human, animal, and environmental health sectors. | **Rapid response:** assess emergency preparedness and response planning, exercising response plans, emergency response operation, linking public health and security authorities, risk communication, access to communications infrastructure, and trade and travel restrictions. | **Health system:** assess health capacity in clinics, hospitals, and community care centers; medical countermeasures and personnel deployment; healthcare access; communications with healthcare workers during a public health emergency; infection control practices and availability of equipment; and capacity to test and approve new countermeasures. | **Compliance with international norms:** assess IHR reporting compliance and disaster risk reduction; cross – border agreements on public health emergency response; international commitments; completion and publication of WHO JEE and the World Organisation for Animal Health (OIE) Performance of Veterinary Services (PVS) Pathway assessments; financing; and commitment to sharing of genetic and biological data and specimens. | **Risk environment:** assess political and security risk; socioeconomic resilience; infrastructure adequacy; environmental risks; and public health vulnerabilities that may affect the ability of a country to prevent, detect, or respond to an epidemic or pandemic and increase the likelihood that disease outbreaks will spill across national borders. |
| **\*Indicators present in only UHC** | **Indicators present in only GHS** | | | **Indicators present in both UHC and GHS** | | |

**Source:** Global Health Security Index 2019 (https://www.ghsindex.org/) and World Health Organization(https://www.who.int/data/gho/data/themes/universal-health-coverage).

many areas of overlap (in immunization, infection diseases, and service capacity areas of UHC indicators and prevention, detection, response, health systems and compliance indicators of GHS) and few divergences (family planning, ANC delivery and NCD services indicators of UHC and some of the prevention, detection, response and Risk environment indicators of GHS). None of the indicators for non-communicable diseases in the UHCI are included in the list of indicators in the GHSI. It is also visible that there are few indicators for UHCI (only 17) compared to GHSI (34 indicators and 85 sub-indicators). Of course, the health security (compliance with the IHR) indicator of UHCI is by itself an index for international health security preparedness, and it consists of the majority of the indicators used to estimate the GHSI.

## Global health security

**The distribution of health security preparedness.** Collectively, global preparedness for health security is quite weak: the average GHSI is 40.2 (out of a possible 100) while it is 51.9 (out of a possible 100) among high-income countries. Table 2 shows that the GHSI is variable

**Table 2. Level of global health security in the six WHO regions, 2019.**

| WHO regions | Global Health Security Index by Category | | | | | | |
|---|---|---|---|---|---|---|---|
| | Average | Prevention | Detection and reporting | Rapid response | Health system | Compliance with international norms | Risk and vulnerability |
| **Africa** | 31% | 25% | 32% | 31% | 15% | 47% | 59% |
| **Americas** | 41% | 35% | 41% | 38% | 25% | 51% | 42% |
| **South-East Asia** | 40% | 35% | 43% | 41% | 27% | 48% | 51% |
| **Europe** | 51% | 49% | 56% | 46% | 40% | 54% | 32% |
| **Eastern Mediterranean** | 35% | 30% | 36% | 36% | 22% | 40% | 53% |
| **Western Pacific** | 38% | 29% | 37% | 39% | 24% | 44% | 41% |
| **Global** | **40%** | **35%** | **42%** | **38%** | **26%** | **49%** | **45%** |

**Source:** Global Health Security Index 2019: https://www.ghsindex.org/.

across the different WHO regions. The index is highest in Europe (51%) and lowest in Africa (31%). Among the categories of the index, the health system capacity has the lowest index (26%) and compliance with international norms has the highest index (49%).

**Countries with a high risk of GHS threats have a low capacity for GHS.** Fig 1 shows that there are countries with a high risk of GHS threats, but have a low capacity for GHS. There is a significant and negative correlation between risk environment (risk and vulnerability) and capacity for GHS (r = -0.604, p<0.001).

## Universal health coverage

Table 3 shows the average UHCI is 66%; it is highest in Americas region (79%), and lowest in African region (46%).

## Global health security and universal health coverage

**There is a moderate and significant relationship between UHC index and GHS index.** There was a significant relationship between UHCI and GHSI (r = 0.662, p <0.001) (see Fig 2). The UHCI alone explains more than 43% of the variability in GHSI. The remaining 57% of the variability is explained by other factors which are not captured by the UHCI.

**Countries with a high risk of GHS threats have a low capacity for UHC.** Countries with a high risk of GHS have a low capacity for UHC. There is a significant and negative correlation between risk environment (risk and vulnerability) and capacity for UHC (r = -0.794, p<0.001). This negative correlation is even stronger than the correlation between risk environment (risk and vulnerability) and capacity for GHS (r = 0.604, p<0.001).

**Global health security from what?.** Fig 3 shows the correlation between the GHSI and the individual component indices of the UHCI (maternal, neonatal and child health index (MNCHI), infectious diseases index (IDI), non-communicable diseases index (NCDI), and service capacity index). The GHSI is moderately and significantly associated with the MNCHI, IDI and service capacity index (r = 0.623 (p<0.001), 0.594 (p<0.001), 0.638 (p<0.001), respectively). However, there is no relationship between the GHSI and the NCDI (r = 0.063, p>0.05).

**Outliers: Countries with discordant relationship between universal health coverage index and global health security index.** Fig 4 shows that the majority of countries with third and fourth quartiles of UHCI (for maternal and child health and infectious diseases services)

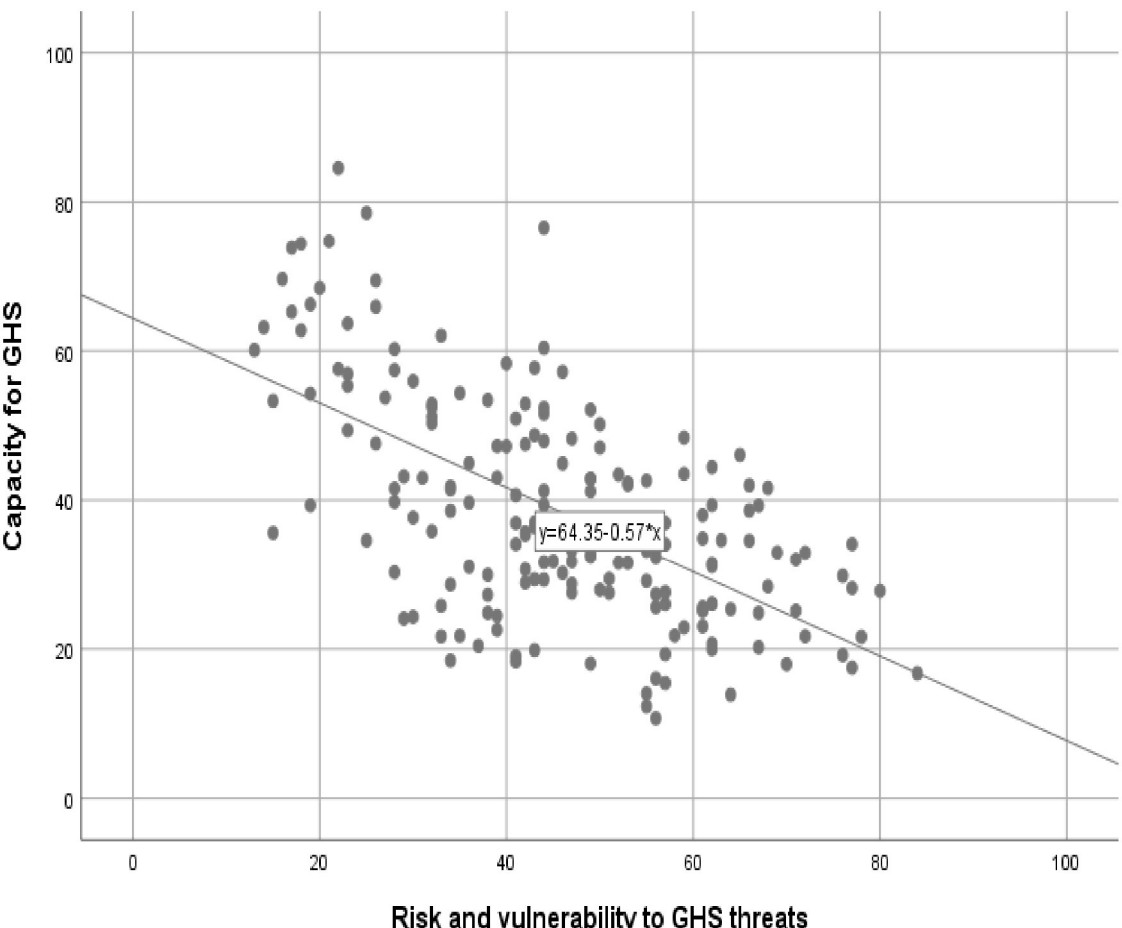

**Fig 1. Scatter plot of risk of GHS threats and capacity for GHS in 183 countries.** GHS: Global Health Security. Source: Global Health Security Index 2019 (https://www.ghsindex.org/).

**Table 3. Level of universal health coverage index in the six WHO regions, 2018, in 183 countries.**

| WHO region | Universal Health Coverage Index | | | | |
|---|---|---|---|---|---|
| | UHC index of service coverage | Reproductive, maternal, newborn and child health | Infectious diseases | Non-communicable diseases | Service capacity and access |
| **Africa** | 46% | 54% | 42% | 71% | 30% |
| **Americas** | 79% | 84% | 72% | 71% | 90% |
| **South-East Asia** | 56% | 71% | 45% | 63% | 50% |
| **Europe** | 77% | 86% | 73% | 61% | 94% |
| **Eastern Mediterranean** | 57% | 66% | 45% | 61% | 60% |
| **Western Pacific** | 77% | 85% | 69% | 65% | 95% |
| **Global** | 66% | 75% | 58% | 65% | 70% |

**Source:** World Health Organization: https://www.who.int/data/gho/data/themes/universal-health-coverage.

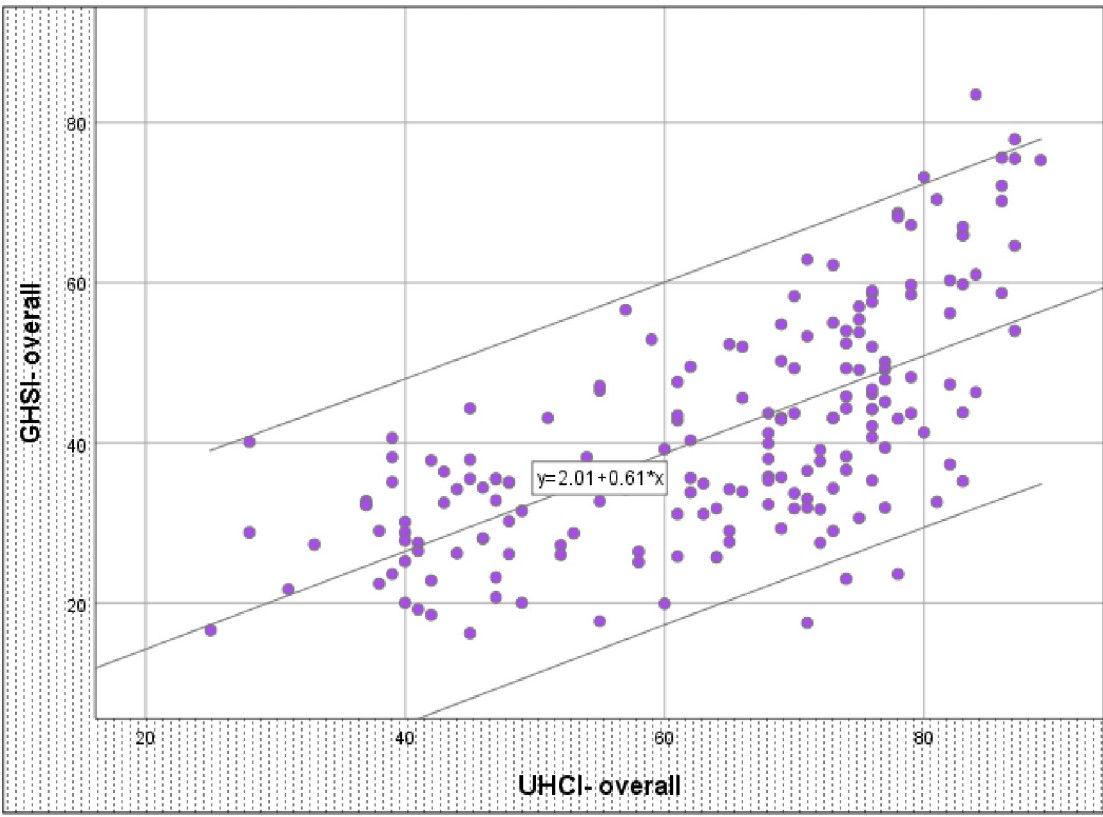

**Fig 2. Scatter plot of universal health coverage index and global health security index in 183 countries.** GHSI: Global Health Security Index; UHCI: Universal Health Coverage Index. Source: Global Health Security Index 2019 (https://www.ghsindex.org/) and World Health Organization(https://www.who.int/data/gho/data/themes/universal-health-coverage).

have above the median GHSI. The figure also shows that majority of countries with the first and second quartiles of UHCI (for maternal and child health and infectious diseases services) have below the median GHSI. Nevertheless, there are also countries with exceptions to these main findings (Fig 4).

There are four groups of countries with a discordant (incongruous) relationship between their UHCI and GHSI: (1) countries (such as Ethiopia, Uganda, Vietnam, India and Indonesia) have GHSI scores above the median but UHCI scores in the first quartile; (2) countries (such as Philippines, Cambodia and Kenya) have GHSI scores above the median but UHCI scores in the second quartile; (3) countries (such as Fiji, Venezuela and Algeria) have GHSI scores below the median but UHCI scores in the third quartile; and (4) countries (such as Cuba and North Korea) have UHCI scores more than the median but GHSI scores less than the median.

## Discussion

This study has identified that there is inadequate global capacity for health security; no country or region is fully prepared for GHS; countries with high risk from GHS threats have lower capacity for GHS and UHC; UHCI is significantly associated with GHSI; however, UHC explains only 43% of the variability in GHSI; and NCDI is not correlated with GHSI. Other studies also reported similar findings. Nirmal K, et al. conveyed that countries vary widely in

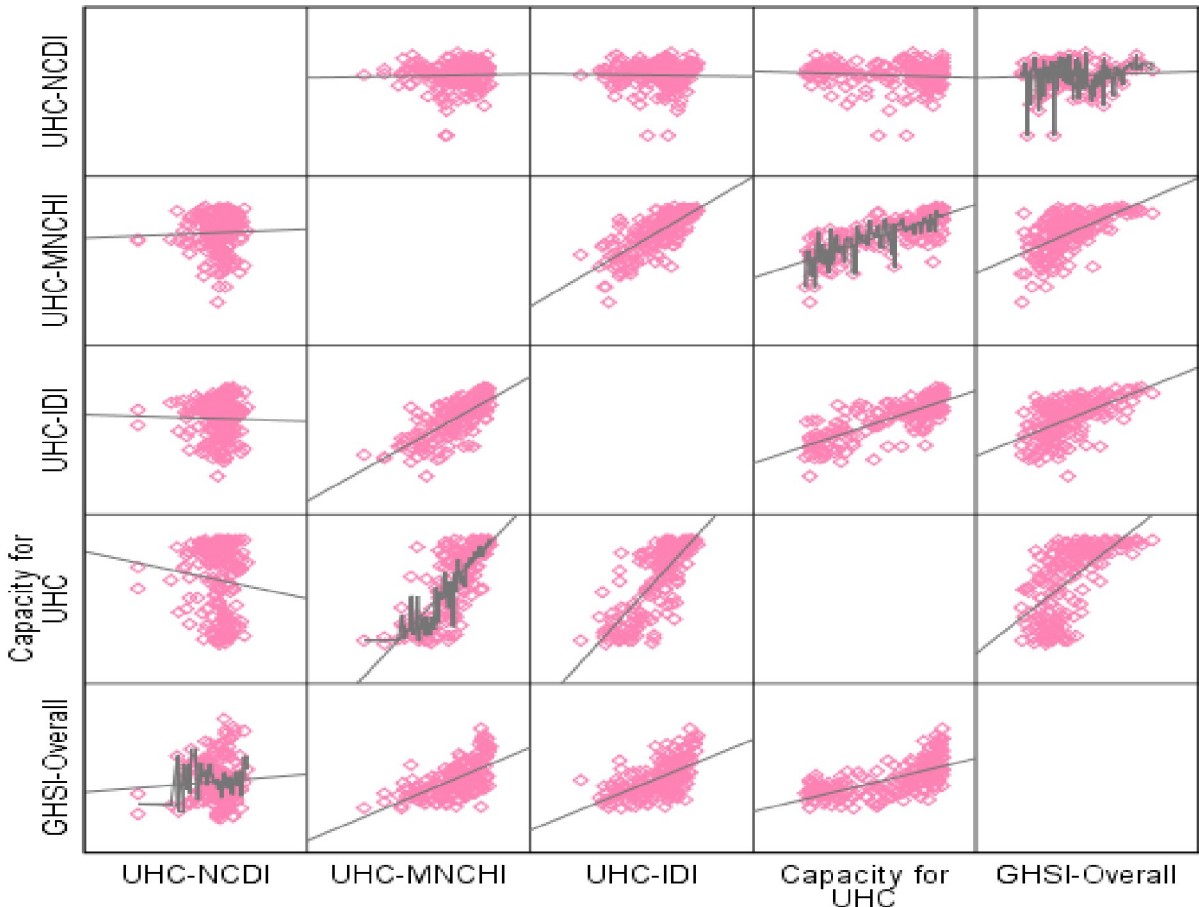

**Fig 3. Scatter plot of sub-indices of universal health coverage index and global health security index in 183 countries.** GHSI: Global Health Security Index; UHCI: Universal Health Coverage Index; MNCHI: maternal, neonatal and child health; ID: infectious diseases; NCD: non-communicable diseases. Source: Global Health Security Index 2019 (https://www.ghsindex.org/) and World Health Organization(https://www.who.int/data/gho/data/themes/universal-health-coverage).

their abilities to prevent, detect, and control new infectious outbreaks. One-third of the 182 countries analysed have limited capacities to prevent, detect and respond to outbreaks of infectious diseases [21]. A study by Oppenheim B, et al. also demonstrates that countries with higher risk and vulnerability have a lower capacity than others. There is overlap of areas with 'low GHS preparedness' and 'high disease emergence risk' [22].

However, risks to health security can easily be transmitted from one place to another; as a result, epidemics can easily become pandemics. The current epidemic of COVID-19 shows that an outbreak in one district of a country will become a national and then global health security issue, indicating that no country is safe until every country is safe [23]. Health security is a regional and global public good, which requires a collective responsibility [24]. Compliance with IHR (2005) is a critical step towards an effective and sustainable response to GHS [8]. Countries with weak capacity should have a motivation for national prioritization to build IHR (2005) core capacities for GHS. These countries also have a strong case for external assistance until they are able to prevent, detect and respond to PHEs. If the IHR (2005) capacity in countries is systematically and proactively coordinated and managed, the benefits of investments in one country will extend to the wider region or the globe [25]. This will enable us to

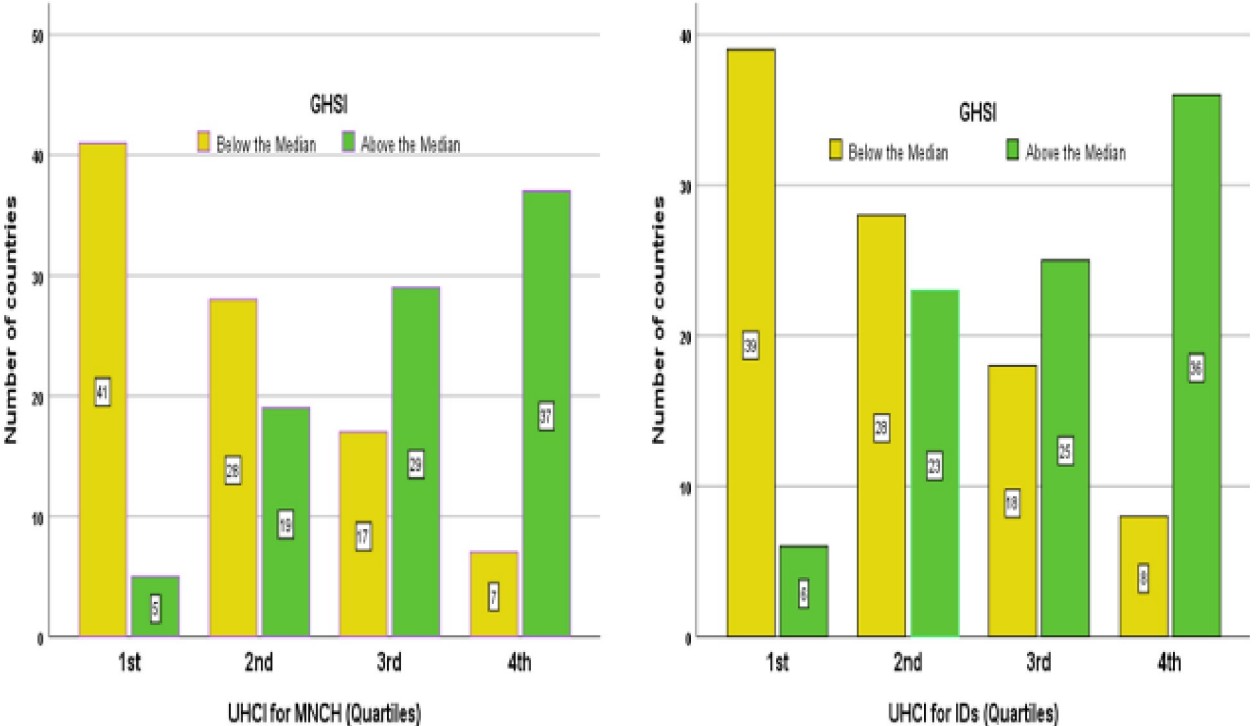

**Fig 4. Countries above and below the median GHS index in the four quartiles of UHC index for maternal and child health and infectious diseases services.** UHCMNCH: Universal Health Coverage Index for maternal, Neonatal and Child Health Services; UHCID: Universal Health Coverage Index for Infectious Diseases; GHSI: Global Health Security Index. Source: Global Health Security Index 2019 (https://www.ghsindex.org/) and World Health Organization(https://www.who.int/data/gho/data/themes/universal-health-coverage).

create a collective capacity which can also be utilized by other countries to contain and mitigate epidemics before they become pandemics. This has the implication that health security requires a regional and global level response, management and coordination, in addition to the national level response [26, 27].

These endeavours require a new financing strategy. It is time that we establish and/ or strengthen a global health security insurance mechanism that pools capacity in both cash and kind to prevent, detect and respond to health emergencies happening anywhere in the globe. It should be generated from different sectors and organizations to support the implementation of the "One Health" approach towards GHS and safer world. The World Bank's Pandemic Emergency Financing Facility is such a mechanism that provides supplemental financing for responding to potential pandemics [28]. However, this World Bank's Pandemic Emergency Financing Facility is struggling to deliver on its innovative promise. Therefore, a reform is required so that it can contribute towards an effective and sustainable GHS response [28].

The moderate and significant correlation between the overall UHCI and GHSI implies that changes in one will lead to changes in the other. Implementation of initiatives for GHS, such as IHR (2005), helps countries to address everyday health challenges from endemic diseases [29]. It is therefore appropriate for global health organizations and countries to undertake more joint and strategic planning with a focus on both UHC and GHS [30]. Global health security is more easily achieved when all people can obtain the health services they need without suffering financial hardship" [31]. The attainment of UHC is crucial to establish a first-line of defence against threats to health security. Initiatives and strategies towards UHC will also

build capacity towards health security, including in the prevention and control of emerging diseases. Inadequate UHC in one country may cause population movements through borders to seek care in neighbouring countries. This will increase the risk for the spread of infections to other countries. There are reports that Ebola also scattered in west Africa due to population movements across borders in search of health services [32].

Universal health coverage can also be a means to strengthen social solidarity and enhance human (health) security [33]. Achievement of the financial protection goal of UHC can be an effective strategy to prevent poverty (by avoiding catastrophic expenditure or impoverishment) which will in turn ensure health security. Thailand was able to reduce annual impoverishment from medical costs from 2·71% to 0·49% in 10 years by providing UHC to its population [34]. This has implications that UHC reduces poverty and inequity, which are linked to national, human and health security [35].

Achieving UHC requires strong health systems, designed according to the primary health care (PHC) approach, which facilitates the provision of a continuous and comprehensive care closer to the community by addressing both the supply and demand sides of health care. This will be possible only if there is a political commitment to enhance the three pillars of PHC: universal access to quality health services, empowered people and communities, and multisectoral policy and action for health [36, 37]. Strong health systems are also vital to achieve health security. The Ebola crisis in west Africa in 2014 occurred in countries with weak health systems and low levels of UHC. The lack of access to primary care fuelled the spread of the epidemic in these countries [38]. The current epidemic of coronavirus disease 2019 (COVID-19) in China demonstrated the benefit of strong health systems to mitigate the impact of epidemics. There were higher case-fatality rates in Hubei (about 2·9% on average), which had weaker health systems capacity, than the other provinces of China (about 0·7% on average). There was even lower case-fatality rates in the most developed provinces, such as Zhejiang (0 deaths among 1,171 cases) [39, 40]. It is, therefore, important that countries strengthen their health systems, through the PHC approach, [41] to achieve both GHS and UHC [42]. This requires a shift away from vertical (and competitive) tensions to horizontal (and collaborative) solutions between these two global health agendas [43]. A diagonal approach (a proactive and balanced approach that concurrently addresses the requirements of specific priorities while providing opportunities for strengthening health systems) [44] in investments and service delivery will systematically and proactively facilitate this shift from tensions to solutions [45]. This requires political commitment and coordination of stakeholders at all levels [46].

This study also identified that there are areas of convergence and divergence between UHCI and GHSI indicators. The divergences are due to indicators related NCDI, which do not have correlation with GHSI, and activities conducted by institutions outside the health sector. Infectious diseases have been the focus of GHS since the launch of health security initiatives at regional and global levels [6]. The GHSA is skewed to highly virulent IDs and bioterrorist threats, but has neglected NCDs though they are the leading causes of morbidity and mortality worldwide [43, 47]. Horton R. also argued that the world has failed to respond effectively to the rising burden of NCDs due to a pervasive fear that has displaced all other health concerns [48].

We contend that addressing NCDs is crucial for an effective and sustainable GHS response. The risks for NCDs, including behaviours (such as smoking and drinking) and products (such as food, cigarettes and drugs), are shared and transferred from place to place and from person to person [49, 50]. NCDs and IDs share common features, such as long-term care needs and overlapping high-risk populations, and have direct interactions due to increased susceptibility to IDs in individuals with NCDs. A systematic review and meta-analysis on the prevalence of comorbidities in patients with Middle East respiratory syndrome coronavirus (MERS-CoV)

indicates that diabetes and hypertension are equally prevalent in approximately 50% of the patients and cardiac diseases and obesity are also present in 30% and 16%, respectively, of the cases. The review recommended that protection against MERS-CoV and other respiratory infections can be improved if public health vaccination strategies are tailored to target persons with chronic disorders [51]. The ongoing outbreak of COVID-19 clearly illustrates the impact of NCDs co-morbidity on the severity of illness and rate of fatal outcome. The results of a single-centre case series showed that patients with severe COVID-19–associated pneumonia, admitted to the ICU, were more likely to have underlying comorbidities [52]. Furthermore, an analysis of 72,314 patient records of confirmed and suspected cases of COVID-19 showed that the case-fatality rate was 5–10 times higher in patients with comorbidities [53].

It is, therefore, relevant and appropriate that Remais JV, et al. advocate for integrated care and surveillance of NCDs and IDs based on the observed convergence of disease burden [54]. Initiatives that advance the prevention and control of NCDs can support the goals of GHS through direct and indirect mechanisms. NCDs prevention and control programs will help reduce the burden of IDs at population level. Moreover, health systems platforms to prevent, detect and respond to NCDs can also be used for the prevention and control of IDs [55]. In addition, reducing the burden of NCDs has a significant positive economic impact that further strengthens the health system for prevention and control of IDs [56]. Therefore, incorporating NCDs with the GHSA efforts can have a synergistic effect towards resilient health systems, which can respond in ordinary and extraordinary times, to ensure human and health security through a multi-sectoral action [35].

The other reason for the divergence between UHCI and GHSI relates to activities outside the health system, such as animal and agriculture sectors. These activities are collectively described as a "One Health" approach to health security [10]. The "One Health" approach supports GHS by improving coordination, collaboration and communication at the human-animal-environment interface to address shared health threats such as zoonotic diseases, antimicrobial resistance and food safety [57].

The GHSA was launched by more than 20 countries, in collaboration with the Centers for Disease Control and Prevention (CDC), the World Health Organization, the World Organisation for Animal Health and the Food and Agriculture Organization, to coordinate action and promote GHS as an international security priority [55]. A GHSA demonstration project in Uganda confirmed substantial improvements to the ability of the country's public health system to detect and respond to health threats [58]. A similar project in Vietnam enhanced emergency operation centre, laboratory system and information systems platform [59]. The IHR (2005) joint external evaluation process is also catalysing dialogue and partnership for a multi-sectoral approach to evaluate preparedness for health security [60, 61]. There are also efforts to expand global health activities in partnership with security and foreign policy though there is a concern for the possibility of over securitization of global health [62, 63]. However, much more needs to be done to effectively address human, animal, and environmental health collectively in order to prevent the spread of infectious diseases of global health concern [20]. Currently, fewer than 30% of countries demonstrate the existence of mechanisms for sharing data for human, animal, and wildlife surveillance. There are also land-use changes in many countries, which need to addressed strategically to prevent the emergence of infectious diseases [64].

Overall, this study emphasises the need for countries to have a strong capacity for both UHC and GHS. There are two groups of countries which require especial attention due to the discrepancies in the relationship between UHCI and GHSI. Countries such as Ethiopia, Uganda, and Vietnam have GHSI score above the median but UHCI score below the first quartile. It can be noted that these countries are supported by the GHSA and have benefited

from it [65]. Other countries such as Cuba and North Korea have UHCI score more than the median but GHSI score less than the median. These countries are isolated (themselves or by others) from the rest of the world [66], and hence, do not consider GHS as a national priority [67]. We argue that no country is safe until every country is safe, and hence no country should not isolate itself from GHS platforms and activities. We recommend further research to explore and explain the discordances between UHCI and GHSI in these countries.

This study has both strengths and limitations. The strengths of this study are: (1) it is the first of its kind, as there is no empirical study that attempts to systematically validate the relationship between UHC and GHS; (2) it is based on two large data sets from almost all countries around the globe; (3) it provides evidence towards a synergistic response for UHC and GHS; and, (4) it shows areas that the UHC and GHS communities should individually and collectively address to fill the gaps towards achieving the health-related SDG and an equitable and safer world. The limitations of the study are: (1) it is based on index measures that in turn are constructed using different data sources, which may have implications for the quality of the data used and the evidence generated from it; (2) UHCI are constructed based on 17 indicators which do not necessarily show the real picture, as certain key indicators (such as skilled delivery service, malaria treatment, TB diagnosis, cancer treatment) are omitted from the index and hence, this study; (3) UHCI and GHSI share a number of data elements such as service capacity, which are included in the construction of the indices; this may have an effect on the correlation analysis; and, (4) the indices are developed using aggregate data, at country-level, which may have weaknesses related to quality such as accuracy and reliability. Nevertheless, we consider that these limitations are not systematically influencing the results, and that they do not affect the analysis and recommendations of this study.

## Conclusion

In conclusion, there is inadequate global preparedness for health security and no country or region is fully prepared for GHS. Effective response to GHS demands a global and regional mechanism that supports and facilitates prevention, detection and response to emergencies at national and regional levels. Moreover, the aspiration for GHS will not be realized without UHC; hence, the tension between GHS and UHC should be transformed into a synergistic planning, financing and implementation, through a diagonal investment and service delivery approach, including differentiated, integrated and community-led services. We argue that strengthening the health systems, in tandem with the principles of PHC (universal access to quality health services, empowered people and communities, and multisectoral policy and action for health), and implementing a "One Health" approach through a multi-sectoral approach will progressively enable countries to realize both UHC and GHS towards a healthier and safer world that everyone aspires to live in.

## Author Contributions

**Conceptualization:** Yibeltal Assefa, Peter S. Hill, Charles F. Gilks, Wim Van Damme, Remco van de Pas, Simon Reid.

**Data curation:** Yibeltal Assefa, Solomon Woldeyohannes, Simon Reid.

**Formal analysis:** Yibeltal Assefa.

**Investigation:** Yibeltal Assefa.

**Methodology:** Yibeltal Assefa.

**Visualization:** Yibeltal Assefa.

**Writing – original draft:** Yibeltal Assefa.

**Writing – review & editing:** Yibeltal Assefa, Peter S. Hill, Charles F. Gilks, Wim Van Damme, Remco van de Pas, Solomon Woldeyohannes, Simon Reid.

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
