## [Decision Letter · Decision Letter 0]

24 Sep 2020

PONE-D-20-11383

Global Health Security and Universal Health Coverage: Understanding Convergences and Divergences for a Synergistic Response

PLOS ONE

Dear Dr. Assefa,

Thank you for submitting your manuscript to PLOS ONE. After careful consideration, we feel that it has merit but does not fully meet PLOS ONE’s publication criteria as it currently stands. Therefore, we invite you to submit a revised version of the manuscript that addresses the points raised during the review process.

Please make sure that all reviewers comments have been clearly addressed or a rebuttal provided. 

We look forward to receiving your revised manuscript.

Kind regards,

Irene Agyepong

Academic Editor

PLOS ONE

Journal Requirements:

"No"

Reviewers' comments:

Reviewer's Responses to Questions

**Comments to the Author**

1. Is the manuscript technically sound, and do the data support the conclusions?

Reviewer #1: Partly

Reviewer #2: Yes

2. Has the statistical analysis been performed appropriately and rigorously? 

Reviewer #1: Yes

Reviewer #2: Yes

3. Have the authors made all data underlying the findings in their manuscript fully available?

Reviewer #1: No

Reviewer #2: Yes

4. Is the manuscript presented in an intelligible fashion and written in standard English?

Reviewer #1: Yes

Reviewer #2: Yes

5. Review Comments to the Author

Reviewer #1: This is a relevant ecological study that involved 183 countries. The study assessed the relationship between GHS and UHC by examining their association using two recent quantitative indices. However, this study should undergo a major review before considering it for publication.

(1) The authors should replace “association” with “relationship”

(2) It was not clear if the authors covered the following areas (First, we will describe the concordance and discordance of the indicators for UHCI and GHSI. Second, we will analyze the level and equity of UHCI and GHSI across the six WHO regions. Third, we will assess the association between UHCI and GHSI) in the statistical analysis section.

(3) The authors should consider applying control-chart to explain the outliers and countries above and below the median GHS index and UHC index for maternal and child health and infectious diseases. Deploying this method will enable the authors to identify special- and common-cause variation.

(4) It will be more relevant if the authors can examine what made some countries to have discordant associations between universal health coverage index and global health security index.

(5) How did the authors manage missing data?

(6) This is an ecological study that used aggregate data at country-level. Thus I will advise the authors to highlight other important weaknesses of this study.

Reviewer #2: This is a relevant study that aimed to assess the relationship between GHS and UHC by determining their association using two recent quantitative indices (GHSI and UHCI). The authors identified the presence of convergence and/or divergence between UHC and GHS. Lastly, they assessed the association between UHCI and GHSI and identified “for whom” and “from what” GHS is designed to provide security.

Key findings from this study include moderate and significant association between GHSI and UHCI, and no association between GHSI and non-communicable diseases.

The authors conclude that the aspiration for GHS will not be realized without UHC; hence, the tension between these two global health agendas should be transformed into a synergistic solution. They further propose a one health approach as a potential synergistic solution.

While this manuscript is relevant and an easy read, it could be further strengthened by addressing the following comments:

1. The first paragraph of the findings should be included in the methods section.

2. For presentation of the results, it is useful to start with an overview of the number of indicators per index before describing concordance and discordance.

3. On page 6, the key categories for measuring GHSI are indicated as prevention, detection, response, health system, compliance, and risk. In table 2 resilience is indicated as a category. Please review and make the necessary correction for consistency.

4. What is the data source for table 2?

5. The visual presentation of all figures needs to be improved.

6. Indicate the data source for all figures and tables as figures and tables must be “stand alone”.

7. How was risk and vulnerability for GHS threats assessed in figure 1?

8. Under section 4.3, the authors report that “The GHSI is most strongly associated with the MNCHI, IDI and service capacity index (r = 0.623, 0.594, 0.638)”. Such values of correlation are more appropriately described as being moderate and not strong.

9. Explain why for figure 4 you choose to analyse the number of countries above and below the median GHS index and UHC index for specifically “maternal and child health and infectious diseases” and not other indices. How does the information in figure 4 link with the text for section 4.4? The heading for section 4.4 seems to suggest a broader analysis not only limited to “maternal and child health and infectious diseases” as indices. Please revise section 4.4 accordingly.

10. There is repetition of points in the first 2 paragraphs of the discussion. Please review the text and summarize as needed to improve readability.

11. The authors discuss that “Countries with weak capacity have a strong case for external

assistance until they are able to prevent, detect and respond to PHEs”. A strong case for national prioritization to detect and respond to PHEs is also very important and must be captured in the statement.

12. The concept of a “one health approach” is introduced in the discussion and discussed in more detail much later in the discussion. The authors need to add a few sentences to explain what this approach means early on.

13. The authors make categorical statements such as “This mechanism will definitely complement the national level health insurance mechanism”. Where is the evidence for this? The authors should be cautious in making such statements without substantiating the point being made- alternatively, such statements can be phrased emphasizing the possibility of expected outcomes as being argued by the authors. There are several of such statements in the discussion and they should be modified accordingly.

14. Please check the discussion for grammatical errors.

15. Please reference and explain what a diagonal approach (as used in the discussion) is e.g. https://www.unicef.org/sowc08/docs/sowc08_panel_2_6.pdf

16. The authors suggest in the discussion that “…. also identified two groups of countries which require especial attention due to the discrepancies in the association between UHCI and GHSI”. The basis for suggesting that these groups of countries require special attention is unfounded. Yes, they need attention but so do the many countries that have poor correlation between UHCI and GHSI. The authors should rephrase the sentence to capture the need for countries to have both a strong UHCI and GHSI.

17. Where are the results of the chi-square and regression analysis in the manuscript?

18. The authors indicate as a limitation that -“UHCI are constructed based on 17 indicators which do not necessarily show the real picture, as certain key indicators might be omitted from the index”. It will be useful to cite examples of these key indicators that may be omitted in text or in an appendix.

19. The authors also cite a limitation of the study as “UHCI and GHSI share a number of data elements such as service capacity, which are included in the construction of the indices; this may have an effect on the correlation analysis”. In what way/how will the correlation analysis be affected due to shared data elements in both UHCI and GHSI?

20. In general, very good points are made and discussed in the discussion section. However, the discussion could benefit from revision to improve its flow, connection of ideas/arguments and grammatical errors.

21. Please review the colour coding for figure 4- usually the red colour is applied where there is cause for concern.

6. PLOS authors have the option to publish the peer review history of their article (what does this mean?). If published, this will include your full peer review and any attached files.

Reviewer #1: No

Reviewer #2: **Yes: **Hannah Brown Amoakoh

---

## [Author Response · Author response to Decision Letter 0]

6 Oct 2020

Point-by-point response to the comments

Reviewer #1: This is a relevant ecological study that involved 183 countries. The study assessed the relationship between GHS and UHC by examining their association using two recent quantitative indices. However, this study should undergo a major review before considering it for publication.

(1) The authors should replace “association” with “relationship”

We accept the comment. We have replaced “association” with “relationship”. 

(2) It was not clear if the authors covered the following areas (First, we will describe the concordance and discordance of the indicators for UHCI and GHSI. Second, we will analyze the level and equity of UHCI and GHSI across the six WHO regions. Third, we will assess the association between UHCI and GHSI) in the statistical analysis section.

We accept the comment. We have provided the data analyses conducted to achieve all of these three objectives (describe the concordance and discordance of the indicators for UHCI and GHSI, analyze the level and equity of UHCI and GHSI across the six WHO regions, and assess the association between UHCI and GHSI). 

(3) The authors should consider applying control-chart to explain the outliers and countries above and below the median GHS index and UHC index for maternal and child health and infectious diseases. Deploying this method will enable the authors to identify special- and common-cause variation.

We accept the comment. We have developed a control-chart (Figure 2) to demonstrate outliers. We have also refined Figure 4 and presented separate figures for maternal and child health and infectious diseases. 

(4) It will be more relevant if the authors can examine what made some countries to have discordant associations between universal health coverage index and global health security index.

We appreciate the comment. We have provided explanations on pages 16 and 17. These explanations may not be sufficient; however, it will require further research to explore and explain the discordances between UHCI and GHSI in these countries. We have recommended this as a future research priority. 

(5) How did the authors manage missing data?

We accept the comment. List-wise deletion is used to manage the missing data.

(6) This is an ecological study that used aggregate data at country-level. Thus I will advise the authors to highlight other important weaknesses of this study.

We accept the comment. We have included quality as an important potential weakness of the aggregate data collected from countries. 

 

Reviewer #2: This is a relevant study that aimed to assess the relationship between GHS and UHC by determining their association using two recent quantitative indices (GHSI and UHCI). The authors identified the presence of convergence and/or divergence between UHC and GHS. Lastly, they assessed the association between UHCI and GHSI and identified “for whom” and “from what” GHS is designed to provide security.

Key findings from this study include moderate and significant association between GHSI and UHCI, and no association between GHSI and non-communicable diseases.

The authors conclude that the aspiration for GHS will not be realized without UHC; hence, the tension between these two global health agendas should be transformed into a synergistic solution. They further propose a one health approach as a potential synergistic solution.

While this manuscript is relevant and an easy read, it could be further strengthened by addressing the following comments:

1. The first paragraph of the findings should be included in the methods section.

We accept the comment. We have moved the first paragraph of the findings to the methods section. 

2. For presentation of the results, it is useful to start with an overview of the number of indicators per index before describing concordance and discordance.

We appreciate the comment. We have presented the indicators for each index in the methods section. The indicators can be presented in the findings section. However, we think that the methods section is more appropriate to present the indicators.

3. On page 6, the key categories for measuring GHSI are indicated as prevention, detection, response, health system, compliance, and risk. In table 2 resilience is indicated as a category. Please review and make the necessary correction for consistency.

We agree that the indicator is a bit confusing. We intended to mean resilience for risk to GHS. We have revised the table so that it repeats the nomenclature used in Global Health Security Index (https://www.ghsindex.org/). 

4. What is the data source for table 2?

Source: Global Health Security Index 2019. https://www.ghsindex.org/.

5. The visual presentation of all figures needs to be improved.

We accept the comment. We have improved the visual presentation of figures.

6. Indicate the data source for all figures and tables as figures and tables must be “stand alone”.

We accept the comment. We have provided data sources for each table and figure. 

7. How was risk and vulnerability for GHS threats assessed in figure 1?

We agree that the indicator is a bit confusing. The Global Health Security Index uses ‘risk and vulnerability’ as its sixth category of indicators. We originally thought that resilience could be translated to mean the opposite of ‘risk and vulnerability’. We have understood from the review that this is confusing. Hence, we preferred to use the original indicator from the source: ‘risk and vulnerability’. 

8. Under section 4.3, the authors report that “The GHSI is most strongly associated with the MNCHI, IDI and service capacity index (r = 0.623, 0.594, 0.638)”. Such values of correlation are more appropriately described as being moderate and not strong.

We accept the comment. We have revised the sentence: The GHSI is moderately and significantly associated with the MNCHI, IDI and service capacity index.

9. Explain why for figure 4 you choose to analyse the number of countries above and below the median GHS index and UHC index for specifically “maternal and child health and infectious diseases” and not other indices. How does the information in figure 4 link with the text for section 4.4? The heading for section 4.4 seems to suggest a broader analysis not only limited to “maternal and child health and infectious diseases” as indices. Please revise section 4.4 accordingly.

We preferred to focus on “maternal and child health and infectious diseases” because these are the sub-indices significantly associated with GHSI in contrast to the index for NCDs. Otherwise, we do not have any other reason. We, of course, also analysed and presented the data for the number of countries above and below the median for overall GHSI and overall UHCI. 

10. There is repetition of points in the first 2 paragraphs of the discussion. Please review the text and summarize as needed to improve readability.

We accept the comment. We have revised the second paragraph to avoid repetition.

11. The authors discuss that “Countries with weak capacity have a strong case for external

assistance until they are able to prevent, detect and respond to PHEs”. A strong case for national prioritization to detect and respond to PHEs is also very important and must be captured in the statement.

We accept the comment. We have revised and expanded the sentence as follows: Countries with weak capacity have a strong case for national prioritization to build core capacities for GHS according to the IHRs (2005). These countries also have a strong case for external assistance until they are able to prevent, detect and respond to PHEs. 

12. The concept of a “one health approach” is introduced in the discussion and discussed in more detail much later in the discussion. The authors need to add a few sentences to explain what this approach means early on.

We accept the comment. We have introduced the concept of “One Health” in the background section. 

13. The authors make categorical statements such as “This mechanism will definitely complement the national level health insurance mechanism”. Where is the evidence for this? The authors should be cautious in making such statements without substantiating the point being made- alternatively, such statements can be phrased emphasizing the possibility of expected outcomes as being argued by the authors. There are several of such statements in the discussion and they should be modified accordingly.

We accept the comment. We have revised the paragraph to avoid such unsubstantiated statements. We have also checked for similar statements in the discussion section and revised them. 

14. Please check the discussion for grammatical errors.

Thank you for the comment. We have reviewed and edited the manuscript. 

15. Please reference and explain what a diagonal approach (as used in the discussion) is e.g. https://www.unicef.org/sowc08/docs/sowc08_panel_2_6.pdf

Thank you for the comment. We have provided a definition for a diagonal approach (a proactive and balanced approach that concurrently addresses the requirements of specific priorities while providing opportunities for strengthening health systems)

16. The authors suggest in the discussion that “…. also identified two groups of countries which require especial attention due to the discrepancies in the association between UHCI and GHSI”. The basis for suggesting that these groups of countries require special attention is unfounded. Yes, they need attention but so do the many countries that have poor correlation between UHCI and GHSI. The authors should rephrase the sentence to capture the need for countries to have both a strong UHCI and GHSI.

We accept the comment. We have revised the paragraph accordingly. 

17. Where are the results of the chi-square and regression analysis in the manuscript?

Thank you for this comment. We initially conducted chi-square and regression analyses. However, we decided to drop the findings from these analyses. We have revised the methods section that chi-square and regression analyses are no more included.

18. The authors indicate as a limitation that -“UHCI are constructed based on 17 indicators which do not necessarily show the real picture, as certain key indicators might be omitted from the index”. It will be useful to cite examples of these key indicators that may be omitted in text or in an appendix.

We accept the comment. We have provided examples of indicators (such as such as skilled delivery service, malaria treatment, TB diagnosis, cancer treatment) which we think should be included in the UHCI. 

19. The authors also cite a limitation of the study as “UHCI and GHSI share a number of data elements such as service capacity, which are included in the construction of the indices; this may have an effect on the correlation analysis”. In what way/how will the correlation analysis be affected due to shared data elements in both UHCI and GHSI?

A number of indicators such as immunization, health systems (human resources, infrastructure and medicines), and those related to IHC capacities are present in both UHC and GHS. This will result in a collinearity effect as same and similar indicators are present in both indices. 

20. In general, very good points are made and discussed in the discussion section. However, the discussion could benefit from revision to improve its flow, connection of ideas/arguments and grammatical errors.

Thank you for the comment. We have reviewed and edited the manuscript. 

21. Please review the colour coding for figure 4- usually the red colour is applied where there is cause for concern.

We accept the comment. Thank you. We have changed the colour coding for figure 4.

---

## [Decision Letter · Decision Letter 1]

24 Nov 2020

PONE-D-20-11383R1

Global Health Security and Universal Health Coverage: Understanding Convergences and Divergences for a Synergistic Response

PLOS ONE

Dear Dr. Assefa,

Thank you for submitting your manuscript to PLOS ONE. After careful consideration, we feel that it has merit but does not fully meet PLOS ONE’s publication criteria as it currently stands. Therefore, we invite you to submit a revised version of the manuscript that addresses the points raised during the review process.

Please make sure you address all the minor revision comments raised by reviewer 1.  

We look forward to receiving your revised manuscript.

Kind regards,

Irene Agyepong

Academic Editor

PLOS ONE

Reviewers' comments:

Reviewer's Responses to Questions

**Comments to the Author**

1. If the authors have adequately addressed your comments raised in a previous round of review and you feel that this manuscript is now acceptable for publication, you may indicate that here to bypass the “Comments to the Author” section, enter your conflict of interest statement in the “Confidential to Editor” section, and submit your "Accept" recommendation.

Reviewer #2: (No Response)

2. Is the manuscript technically sound, and do the data support the conclusions?

Reviewer #2: Yes

3. Has the statistical analysis been performed appropriately and rigorously? 

Reviewer #2: Yes

4. Have the authors made all data underlying the findings in their manuscript fully available?

Reviewer #2: Yes

5. Is the manuscript presented in an intelligible fashion and written in standard English?

Reviewer #2: Yes

6. Review Comments to the Author

Reviewer #2: Thank you for addressing the comments raised earlier about your manuscript. The manuscript is more clearer now. However, the manuscript could be further improved by addressing the following comments:

1. Include the p-values for all correlation coefficients in the results section to enable readers assess the significance of the associations.

2. You state that the primary health approach is useful in strengthening health systems. It will be useful to remind readers what the basic idea of PHC is and how it relates to strengthening health systems.

3. Under strengths and limitations, you still indicate this …”(3) it uses different analytical approaches (regression analysis and chi-squared statistics) to triangulate the findings of the study;…”. Please remove this statement as you did not perform this analysis.

4. Under the author summary box :What did the researchers do and find? • “We determined the relationship between GHS and UHC by using two recent quantitative indices: UHC index (UHCI) [1, 2] 1,2 and GHS index (GHSI). We conducted correlation and regression analyses”. Please remove the regression analysis as this was not done. Also include the p-values for the results section in the box as you indicate some significant correlations exist.

5. Add p-values to the correlation coefficients in the abstract.

6. In the conclusion you state “…hence, the tension between GHS and UHC should be transformed into a synergistic planning, financing and implementation, through a diagonal investment and service delivery approach”. Explain the term service delivery approach.

7. PLOS authors have the option to publish the peer review history of their article (what does this mean?). If published, this will include your full peer review and any attached files.

Reviewer #2: **Yes: **Hannah Brown Amoakoh

---

## [Author Response · Author response to Decision Letter 1]

27 Nov 2020

Dear Dr Irene,

Thank you for giving us the opportunity to improve our manuscript. The comments from the reviewer are very useful and constructive. 

We are pleased to submit the third version of our manuscript to PLOS ONE. Following is our point-by-point response to the comments from the reviewers. The comments from the reviewers are shown in Calibri and Italics; our responses are in Calibri, Normal and Bold. 

Kind regards,

Dr Yibeltal Assefa 

Corresponding author

Point-by-point response to the comments

Review Comments to the Author

Reviewer #2: Thank you for addressing the comments raised earlier about your manuscript. The manuscript is more clearer now. However, the manuscript could be further improved by addressing the following comments:

1. Include the p-values for all correlation coefficients in the results section to enable readers assess the significance of the associations.

We accept the comment. Thank you. The p-values for all correlation coefficients are provided included in the results section.

2. You state that the primary health approach is useful in strengthening health systems. It will be useful to remind readers what the basic idea of PHC is and how it relates to strengthening health systems.

We accept the comment. Thank you. We have provided the basic idea and pillars of PHC as follows: The PHC approach facilitates the provision of a continuous and comprehensive care closer to the community by addressing both the supply and demand sides of health care. This will be possible only if there is a political commitment to enhance the three pillars of PHC: universal access to quality health services, empowered people and communities, and multisectoral policy and action for health.

3. Under strengths and limitations, you still indicate this …”(3) it uses different analytical approaches (regression analysis and chi-squared statistics) to triangulate the findings of the study;…”. Please remove this statement as you did not perform this analysis.

We accept the comment. Thank you. We have removed the statement. 

4. Under the author summary box :What did the researchers do and find? • “We determined the relationship between GHS and UHC by using two recent quantitative indices: UHC index (UHCI) [1, 2] 1,2 and GHS index (GHSI). We conducted correlation and regression analyses”. Please remove the regression analysis as this was not done. Also include the p-values for the results section in the box as you indicate some significant correlations exist.

We accept the comment. Thank you. We have removed regression analysis. The p-values for all correlation coefficients are provided.

5. Add p-values to the correlation coefficients in the abstract.

We accept the comment. Thank you. The p-values for all correlation coefficients are provided.

6. In the conclusion you state “…hence, the tension between GHS and UHC should be transformed into a synergistic planning, financing and implementation, through a diagonal investment and service delivery approach”. Explain the term service delivery approach.

We accept the comment. Thank you. We have revised it as follows: service delivery approach, including differentiated, integrated and community-led services.

---

## [Editor Report · Decision Letter 2]

14 Dec 2020

Global Health Security and Universal Health Coverage: Understanding Convergences and Divergences for a Synergistic Response

PONE-D-20-11383R2

Dear Dr. Assefa,

We’re pleased to inform you that your manuscript has been judged scientifically suitable for publication and will be formally accepted for publication once it meets all outstanding technical requirements.

Kind regards,

Irene Agyepong

Academic Editor

PLOS ONE
---

## [Editor Report · Acceptance letter]

17 Dec 2020

PONE-D-20-11383R2 

Global Health Security and Universal Health Coverage: Understanding Convergences and Divergences for a Synergistic Response 

Dear Dr. Assefa:

I'm pleased to inform you that your manuscript has been deemed suitable for publication in PLOS ONE. Congratulations! Your manuscript is now with our production department. 

Kind regards, 

on behalf of

Dr. Irene Agyepong 

Academic Editor

PLOS ONE